# Assessment and optimization plan for enhancing public medical laboratory services in the Kyrgyz Republic

Taalaigul Sabyrbekova*[1], Elmira Turkmenova[2], Ping Ling Yeoh[3], Olga Slobodskaya[4], Cebele Wong[1], Brian Chin[1]

1 Asian Development Bank, Manila, Philippines, 2 Ministry of Health, Bishkek, Kyrgyz Republic, 3 Mediconsult Sdn. Bhd., Ampang, Malaysia, 4 Independent consultant for laboratory services, Leiden, Netherlands

* sabyrbekova2011@gmail.com

## Abstract

Middle-income and especially low-income countries continuously seek ways to efficiently utilize their public funds to provide access to quality healthcare for their population. The government of the Kyrgyz Republic is embarking on a *Strengthening Regional Health Security Project* for the optimization and centralization of medical laboratories, which will be implemented as a pilot in two major cities and two regions of the country to improve essential laboratory testing for disease surveillance and diagnostics. Representative state-funded laboratories (6 bacteriological public health laboratories and 11 clinical diagnostic laboratories) were assessed using the World Health Organization's Laboratory Assessment Tool that covered the provision of services, condition of facilities, equipment, financing, reagent procurement, workflow, regulatory documents, and quality control and assurance. The five greatest weaknesses across the laboratories were identified as: (i) weak organization and management of laboratory, (ii) insufficient documentation management, (iii) inadequate facilities for biosafety, (iv) lack of biorisk management, and (v) unmet public health functions. The results of the assessment informed the design and development of a laboratory optimization master plan, including procurement of equipment and reagents, professional development of laboratory staff, and establishing laboratory networks starting with selected laboratories in two regions of the country. The restructuring, upgrading, and continuous quality improvement of laboratory networks will be achieved through strengthening governance capacity for the national laboratory system including regulation, standards, planning, financing, management, monitoring, and studies for innovative solutions. The approach to assessing the current situation and creating an improvement plan for optimizing laboratory services may be useful to countries facing similar challenges.

**Data availability statement:** All relevant data are within the paper. Questions and clarifications on the data may be sent to the corresponding author at sabyrbekova2011@gmail.com.

**Funding:** Research reported in this publication was supported by the Asian Development Bank (ADB) under TA-6818. The content is solely the responsibility of the authors. The findings, interpretations, and conclusions expressed do not necessarily reflect the views of ADB, its Board of Governors, or the governments they represent. Any designation of or reference to a particular territory or geographic area, or use of the term "country" is not intended to make any judgments as to the legal or other status of any territory or area. The funder had no role in study design, data collection and analysis, decision to publish, or preparation of the manuscript.

**Competing interests:** The authors have declared that no competing interests exist.

## Introduction

The Kyrgyz Republic is a country in Central Asia with a population of 7.0 million. In the first decade since independence (1991) there was a steady decline in government healthcare spending, which constituted in 2019 only 2.3% of gross domestic product [1,2]. Since 1991, the Kyrgyz Republic has undertaken several far-reaching health reform programs. The current Healthy People – Prosperous Country Program (2019–2030) guides the country's health sector reforms [3]. This program includes measures to centralize and improve the quality of medical laboratories [4].

The need to optimize the national medical laboratory system was recognized in the Kyrgyz Republic more than a decade ago. In 2014, the Coordination Laboratory Council (CLC) of the Ministry of Health of the Kyrgyz Republic (MOH) was established with the support of World Health Organization (WHO) *Better Labs for Better Health* initiative [5]. CLC consists of national laboratory experts (usually heads of the leading laboratories) and is a counseling, advisory, and coordinating body on laboratory services development [6]. CLC developed a national laboratory policy and strategic plan for 2016–2025 with the goal to improve access to quality laboratory diagnostics, which were approved by the MOH in 2016 [7]. The commitment to providing essential diagnostics [8–10] to the population has been reinvigorated after the COVID-19 pandemic.

The medical laboratory system of the Kyrgyz Republic under the jurisdiction of the MOH consists of more than 300 laboratories, which form separate laboratory networks. Healthcare laboratories belonging to other ministries, such as the Ministry of the Interior or Ministry of Transport and Communications, are also governed by MOH regulations. The composition of laboratory services in the Kyrgyz Republic is shown in Table 1.

The specific roles and functions of the laboratories include the following:

(i)  Public health laboratories perform tests on environment, food, and consumer products safety. They belong to the State Sanitary and Epidemiological Surveillance (SSES) system of the MOH and form a three-tiered hierarchical structure with the national SSES Department at the top, and regional and district centers below. SSES laboratories also examine patient samples for pathogens (virology, bacteriology, parasitology). Confirmatory testing for monitored pathogens and investigation of outbreaks is a core task of SSES laboratories.

(ii)  Clinical diagnostic laboratories (CDLs) form another, much weaker network with fewer horizontal and vertical connections. These laboratories are part of in- or outpatient medical facilities, such as tertiary (major national hospitals and centers), secondary (regional and district hospitals), or primary (Family Medicine Centers – FMCs) facilities. Administratively, these laboratories are responsible to the respective medical facility directors. They perform general clinical, hematology, and biochemical tests.

(iii)  Public laboratories testing TB, HIV/AIDS, and several highly dangerous infections form separate surveillance networks, and are usually structural subdivision of specialized in- or outpatient medical facilities.

**Table 1. Organization of health laboratory services in the Kyrgyz Republic.**

| Level | SSES | CDL | TB | HIV | RCQED | Private | Others |
|---|---|---|---|---|---|---|---|
| National | SSES reference laboratory | CDRL in public hospitals | TB reference laboratory | HIV reference laboratory | Laboratories of RCQED | Private central laboratory | Laboratories of ministries, companies |
| Region | SSES laboratory | CDL in public hospitals, PHC | Regional TB laboratory | Regional HIV laboratory | Anti-plague stations | Private laboratories | As applicable |
| District | SSES laboratory | | District TB laboratory | District HIV laboratory | | | As applicable |
| Council | | PHC centers | | | | Private clinics | |
| Total | 60 | 136 | 17 | 33 | 5 | 50 | 23 |

CDL = clinical diagnostic laboratory, CDRL = clinical diagnostic reference laboratory, HIV = human immunodeficiency virus, PHC = primary health care, RCQED = Republican Center of Quarantine and Especially Dangerous Infections, SSES = State Sanitary and Epidemiological Surveillance, TB = tuberculosis.

(iv) Private sector diagnostic testing is growing quickly. In 2022, it consisted of 9 leading private players, each forming their own networks of laboratories and sample collection points. Private laboratories are licensed by the MOH Directorate of Licensing.

State-funded laboratories do not have a separate budget within their parent organizations. Due to low public funding of healthcare [2], management of medical facilities has a difficult task balancing needs of patient treatment with needs of laboratories. Many objectives of the strategic plan for the development of laboratory services prepared in 2016 [7] have not been implemented yet mainly because of insufficient funding of laboratories. In the Kyrgyz Republic, patients can request laboratory examinations with or without a doctor's referral. In the state-funded CDLs, all tests are free of payment for citizens who are assigned to a medical facility and have a test request from a physician. Unassigned citizens or clients who want to be tested without a doctor's request pay an out-of-pocket fee according to the published MOH Unified Price List [11].

Between 2015 and 2024, three donor-funded projects on laboratory services optimization have been carried out. In 2015, the Swiss Agency for Development and Cooperation (SDC) supported optimization of CDL network, which was conducted as a pilot in the Issyk-Kul region and identified the areas to be prioritized: (i) updating clinical practice guidelines with rational use of laboratory tests, (ii) defining logistics of sample transportation, and (iii) calculating realistic costs of laboratory tests and allocation of an adequate budget for laboratory services [12]. Since 2016, the WHO initiative *Better Labs for Better Health* has been working in the Kyrgyz Republic on laboratory system strengthening. With the initiative's support, two laboratories have achieved ISO 15189 accreditation [13]. In 2019–2024, to strengthen tuberculosis (TB) diagnostics, the United States Agency for International Development (USAID) supported introduction of highly effective methods, quality management system (QMS), system for transportation of samples from district to national level, and information management system in national TB laboratories [14,15].

In 2018, the WHO *Better Labs for Better Health* initiative conducted assessment of public health bacteriology laboratories in Chui region and found only partially implemented QMS. While the necessary documents were available, their management was weak, staffing was insufficient, facilities were in poor condition, biosafety systems were missing, centralized procurement was difficult, and funding was insufficient [16].

In 2021, the MOH has taken several measures to optimize public health SSES laboratories. As a result, laboratories were merged and consolidated into inter-district laboratories. The nomenclature of tests, equipment inventory, and norms of time for conducting laboratory tests by laboratory levels were also standardized [17,18]. The merger of territorial hospitals and FMCs in the districts also resulted in the merger of smaller CDLs. In the capital Bishkek, primary health care

optimization was carried out, resulting in 19 FMCs being consolidated into 10 FMCs [19]. However, the goal of converting some city CDLs into sample collection points has not yet been achieved as additional budget for this has not been yet provided.

The COVID-19 pandemic exposed the problems of medical laboratories in the Kyrgyz Republic, and while it delayed the implementation of the 2030 Program plans, it encouraged MOH to seek technical and financial support to streamline and centralize laboratories as quickly as possible to improve the availability and quality of laboratory services. The Asian Development Bank (ADB) provided such assistance in the form of a five-year *Strengthening Regional Health Security Project* [20,21].

The first phase of the ADB project was this study with the purpose of characterizing the current state of diagnostics and surveillance provided by health laboratories in the country and proposing a plan for their optimization. This was conducted through the assessment of a representative set of medical laboratories in Chui and Osh regions of the country. The approach and results presented here may help other countries in similar settings to assess and improve the quality of laboratory diagnostics and surveillance and increase access to laboratory services.

## Materials and methods

### Selection of representative laboratories

The laboratories were selected by MOH in consultation with CLC and ADB in the two most populated regions of the country, Chui and Osh regions, with around 3.7 million inhabitants, more than a half of the country's population. The capital city Bishkek, the biggest city with a population of approximately 1 million, is in the Chui region. Osh and Chui regions are characterized by high internal migration, and the cities of Osh and Bishkek have the largest specialized hospitals where patients from all over the country are referred to. The selected cities and regions are located close to the borders with neighboring countries where a large flow of people and trade routes take place; thus, they are important for transmissible diseases surveillance. Other criteria for selection were the population coverage, volume of testing and the MOH plans for optimization and consolidation of health care organizations and laboratories [18,22]. These plans envisage centralization and consolidation of laboratories equipped with high-tech equipment, converting small laboratories into sample collection points, and creating a sample transportation system with subsequent digital transmission of test results.

11 CDLs and 6 SSES laboratories were selected for further study and inclusion in the project. Considering the growing concern about the spread of infections with antimicrobial resistance (AMR), among the selected laboratories, all SSES laboratories and 5 CDLs performed testing for AMR and have a potential to develop and broaden this type of testing.

### Assessment of laboratories

Assessment of the current state of health laboratories in general and specifically of the selected laboratories was conducted using the following three approaches.

1. **Review of reports submitted to the MOH.** We analyzed reports of ongoing projects aimed at the improvement of CDLs [23], data from assessments of SARS-CoV-2-testing laboratories [24], and data from the national e-Health Center on the volume of laboratory tests performed and population coverage. Laboratory productivity was determined by the following indicators: number of samples collected per day, number of laboratory tests per day, spectrum of laboratory tests per day, availability of express testing for intensive care units, and 24-hour laboratory monitoring of critically ill patients.

2. **Audit using WHO Laboratory Assessment Tool (LAT) and other tools.** After review of aforementioned reports and data, the selected laboratories were visited by different experts for data collection and audit. The information was gathered about the current state of the facilities, available equipment, ongoing laboratory tests, personnel, and laboratory finances. The annual budget of laboratories for 2021 was calculated as a sum of expenditures on salaries and social

taxes, utilities, other used services, reagents, and equipment. These figures were provided by bookkeeping units of the parent organizations. Information about equipment was gathered from equipment registers and a visual inspection of equipment's condition and whether it is being used. The number and nomenclature of performed tests were taken from the laboratories' reports submitted to MOH. Additionally, some indicators were calculated based on these collected data. The average cost of a test performed by the laboratory was calculated by dividing the annual budget by the number of tests in this year. The average number of tests performed by a laboratory employee in one month was calculated by dividing the monthly number of performed tests by number of employees.

Extensive laboratory audits were performed using the WHO LAT in accordance with WHO guidelines [25]. These audits were carried out in one-day randomized visits and consisted of interviews with organization and laboratory management and laboratory staff, review of representative laboratory documentation, inspection of facilities, and direct observation of routine workflow. For each laboratory, information collected from the audit and above-mentioned sources was entered in the LAT questionnaire. LAT generated scores for 11 aspects of laboratory performance in a range from 0% (in a very poor state, absent) to 100% (in a good state, all requirements are met).

3. **Review of the range of tests offered and of external quality assessment (EQA) of the testing**. An approach to assessing the quality of laboratory services is to evaluate both the range of laboratory tests performed and the extent to which these tests are subject to EQA by a qualified independent EQA program. In this study, we created a standardized list of essential tests and assessed how many of the tests from the list were conducted by each CDL and which proportion of these tests was covered by EQA participation. This provided insight not only into the tests' availability but also the extent of compliance with quality standards across laboratories.

## Results

### Overview of the existing medical laboratories

Selected laboratories represent the whole spectrum of Kyrgyz medical laboratories with different catchment population, specialization, types, and number of performed tests and other characteristics (Table 2).

The volume of testing in selected laboratories differs, sometimes remarkably. Actual numbers of laboratory tests performed might be similar in a national hospital and a General Medical Practice Center (GMPC) (compare CDLs 1 and 8 in Table 2). It should be noted that CDLs also serve patients of ambulatory, outpatient, and departments of hospitals and GMPCs. In some cases, the numbers of ambulatory visits surpass the numbers of hospitalized patients. The testing volume reflects the capacity of the parent medical facility, the range of available tests and the extent of testing automatization. For instance, the CDL of the National Center for Maternal and Child Health showed one of the highest testing volumes due to the combination of these factors. Some laboratories performing bacteriology testing, both SSES and CDL, examine not only patient samples but also food and environmental samples and test for the sterility of instruments and surfaces (SSES 3, 5 and 6, CDLs 3, 5, 9 and 10).

Despite the limitations of the collected data, it gives an idea of how diverse current Kyrgyz laboratories are and an overall picture of each selected laboratory. It also suggests that the biggest laboratories with more personnel and more tests performed show higher efficiency (ratio of tests to personnel, which implies automatization) and the lower cost of the tests (ratio of budget to number of tests).

### Provision of services

From the annual reports of laboratories to the MOH, we compiled a minimal set of tests that were offered by all CDLs. This set consisted of some types of blood count, urinalysis, and tests for glucose, liver enzymes and for some infections (e.g., viral hepatitis, HIV infection, syphilis). When laboratories had an extended test list, it might contain additional tests

**Table 2. General characteristics of the laboratories in 2021.**

| | City | Region | Annual lab budget (KGZ som) | N of tests per month | N of test types | Average cost of test (KGZ som) | N of employees | Tests per employee per month | N of beds | N of hospital admissions per year | N of ambulatory patients per year | Catchment population |
|---|---|---|---|---|---|---|---|---|---|---|---|---|
| **SSES laboratories** | | | | | | | | | | | | |
| 1 National reference laboratory | Bishkek | Chui | 5,532,529 | 565 | 10 | 817 | 6 | 94 | NA | NA | NA | 2,049,100 |
| 2 Bacteriology laboratory | Tokmok | Chui | 1,909,438 | 536 | 19 | 297 | 6 | 89 | NA | NA | NA | 71,400 |
| 3 Subnational reference laboratory | Osh | Osh | 5,270,252 | 6,328 | 22 | 69 | 17 | 372 | NA | NA | NA | 1,684,674 |
| 4 Bacteriology laboratory | Alai | Osh | 1,241,619 | 259 | 11 | 399 | 4 | 65 | NA | NA | NA | 291,969 |
| 5 Bacteriology laboratory 6 Bacteriology laboratory | Nookat* Aravan | Osh Osh | 2,755,306 | 1,802 | 22 | 127 | 16 | 113 | NA | NA | NA | 440,300 |
| **CDLs of the following medical facilities** | | | | | | | | | | | | |
| 1 National Hospital (national reference lab) | Bishkek | Chui | 12,498,523 | 42,011 | 62 | 25 | 38 | 1,106 | 1,070 | 32,381 | 92,671 | 6,636,800 |
| 2 City Clinical Hospital No. 1 | Bishkek | Chui | 11,556,186 | 12,592 | 39 | 76 | 13 | 969 | 294 | 9,218 | 3,348# | 267,200 |
| 3 Republican Infectious Clinical Hospital** | Bishkek | Chui | 22,475,635 | 37,703 | 64 | 50 | 26 | 1,450 | 500 | 9,854 | 11,803 | 2,049,100 |
| 4 National Center for Maternal and Child Health | Bishkek | Chui | 13,984,540 | 85,016 | 74 | 14 | 40 | 2,125 | 684 | 30,061 | 98,249 | 2,408,979 |
| 5 GMPC** | Jaiyl | Chui | 9,733,520 | 19,153 | 87 | 42 | 35 | 547 | 330 | 12,564 | 1,765 | 112,200 |
| 6 GMPC | Tokmok | Chui | 3,492,533 | 42,939 | 48 | 7 | 22 | 1,952 | 421 | 8,226 | No data | 71,400 |
| 7 GMPC (express laboratory) | Panfilov district | Chui | 2,957,558 | 4,539 | 14 | 54 | 9 | 504 | 85 | 2,989 | No data | 47,900 |
| 8 GMPC | Issyk-Ata district | Chui | 8,084,584 | 10,550 | 19 | 64 | 27 | 391 | 200 | 7,262 | 11,622 | 154,300 |
| 9 Osh Interregional Clinical Hospital (subnational reference lab)** | Osh | Osh | 12,870,441 | 31,972 | 58 | 34 | 47 | 680 | 946 | 37,029 | 4,731 | 3,522,700 |
| 10 GMPC** | Kara-Suu | Osh | 6,468,998 | 20,638 | 54 | 26 | 65 | 318 | 500 | 19,508 | 7,367 | 448,600 |
| 11 GMPC** | Uzgen | Osh | 1,147,845 | 16,377 | 12 | 6 | 25 | 655 | 380 | 12,563 | 2,547 | 283,000 |

CDL – clinical diagnostic laboratory, GMPC – General Medical Practice Center, KGS som – Kyrgyz Som, N – number, NA – not applicable, SSES – State Sanitary and Epidemiological Surveillance.

*SSES bacteriology laboratories in Nookat (#5) and Aravan (#6) were merged administratively in 2021. Aravan laboratory is now a subsidiary of Nookat laboratory.

**CDLs performing bacteriology testing, including antimicrobial resistance (AMR) testing.

#This number is from 2019.

Gray shaded cells denote a national or subnational reference laboratory.

for clinical chemistry (e.g., some parameters of lipid profile, electrolytes, inflammation markers), hematology (coagulation), markers of endocrine disorders, and other tests for infections (e.g., herpes virus, protozoa). Bacteriology tests in CDLs and SSES laboratories included identification of pathogens of respiratory, gastrointestinal and systemic infections and their AMR, performed by culturing and microscopy, the range of testing being determined by availability of reagents.

Laboratories report testing performed according to the national test nomenclature, which currently has over 1,000 test types. The list is in some respects redundant, but in other cases not detailed enough. For instance, blood count measurements can be reported under 17 different entries depending on a measured parameter and method. On the other hand, details of AMR testing results – which pathogen was identified to be resistant to which antimicrobial compound – are reported only to the test requester, these data are not collected by MOH and not analyzed for surveillance.

The project proposed a unified list of essential tests in consultation with national experts. The list was based on the tests routinely requested by physicians, already performed in the laboratories and the WHO Model List of Essential In Vitro Diagnostics (EDL) [26], and it consisted of 61 tests. 44 of these tests belong to WHO EDL and represent general clinical chemistry (23 of 26 in EDL), hematology, microbiology, and a few other testing categories. The additional 17 tests targeted diseases that are considered a burden in the country and female-specific conditions (inflammation markers, some cancers, anemia, and thyroid function). This kind of standardization of the essential diagnostics across laboratories should allow using identical testing methods and applying the same quality control and quality assurance.

Using the proposed list of essential tests, we counted how many test types from this list are being performed by each laboratory and for which of those tests laboratories participate in EQA programs. We excluded from our analysis two microbiology tests on the list and consequently bacteriology SSES laboratories because this kind of analysis would require more detailed, currently nonexistent, reporting about identification of bacterial pathogens and their AMR. CDLs, on average, offered 44% of the tests from the list and 29% of the female-specific tests (Table 3). The largest variety of tests were conducted by the large national hospitals: 35–36 tests out of 59 total and 8–9 out of 19 female-specific tests. Participation in EQA was also the highest in the CDLs of the big Bishkek hospitals, but other CDLs did not participate in any EQA, bringing average numbers down to a poor rating of 12–13%.

## Quality of laboratory performance

The quality of laboratory performance was assessed using LAT scores for 11 elements. Not one laboratory scored sufficiently well on LAT, with all elements or average of all elements scoring equal or higher than 85%. Only four laboratories that have been mentored towards ISO accreditation under the WHO *Better Labs for Better Health* initiative or received support from other international projects showed average scores for 10 elements close to or higher than 60%.

We calculated average scores for the laboratories in each network, CDLs and SSES (Fig 1). There was a large variation in each aspect of laboratory performance within both networks. It showed that adhering to standards currently accepted in the international laboratory community is not a condition for a laboratory to operate. Whether or not a laboratory implements some of these standards correlates with the involvement of the laboratory or laboratory head in internationally supported programs or similar professional activities.

Five elements were identified as urgently needing improvement: (i) laboratory organization and management; (ii) documentation; (iii) facilities not meeting biosafety standards; (iv) biorisk management; and (v) poor involvement of CDLs in public health functions.

The following five elements were in better shape but still need improvement: (i) specimen collection, handling and transportation; (ii) data and information management; (iii) equipment including inventory and maintenance of equipment; (iv) reliability (quality) of laboratory tests; and (v) human resources.

**Table 3. Tests offered by the Project CDLs.**

| CDL of medical facility | | Total tests = 59* | | Female-specific tests = 19** | |
|---|---|---|---|---|---|
| | | % | % EQA | % | % EQA |
| 1 | National Hospital (national reference lab), Bishkek | 59% | 0% | 42% | 0% |
| 2 | City Clinical Hospital No. 1, Bishkek | 49% | 41% | 21% | 50% |
| 3 | Republican Infectious Clinical Hospital, Bishkek | 51% | 40% | 16% | 33% |
| 4 | National Center for Maternal and Child Health, Bishkek | 61% | 65% | 47% | 50% |
| 5 | GMPC, Jaiyl | 54% | 0% | 37% | 0% |
| 6 | GMPC, Tokmok | 49% | 0% | 42% | 0% |
| 7 | GMPC (express laboratory), Panfilov district | 37% | 0% | 11% | 0% |
| 8 | GMPC, Issyk-Ata district | 20% | 0% | 21% | 0% |
| 9 | Osh Interregional Clinical Hospital (subnational reference lab) | 46% | 0% | 32% | 0% |
| 10 | GMPC, Kara-Suu | 47% | 0% | 32% | 0% |
| 11 | GMPC, Uzgen | 12% | 0% | 16% | 0% |
| **Average of CDLs** | | **44%** | **13%** | **29%** | **12%** |

CDL – clinical diagnostic laboratory, EQA = external quality assessment, GMPC – General Medical Practice Center.

% – percentage of offered test types out of proposed list, total or female-specific; % EQA – percentage of performed test types that participated in EQA in 2021.

\* – number of test types in the proposed list of essential tests.

\*\* – number of test types from the proposed list that are targeting female-specific conditions.

## Organization and management

As seen in Fig 1, the average score for this component was 40% (47% for SESS laboratories and 36% for CDLs), indicating significant shortcomings in several areas. These include ineffective communication with clients, weak management of laboratory processes and personnel, inadequate financial planning and insufficient implementation of external quality control and assurance systems.

## Documents

The average score for documentation was 37% (44% for SESS and 34% for CDL laboratories). Generally, there is either a complete absence or inadequacy of written procedures for laboratory processes, compromising standardization of testing. Furthermore, most laboratories lack a documentation management system, including mechanisms for regular review, updates, and tracking of documents.

## Facilities

Laboratory facilities scored 46% for SESS and 52% for CDL laboratories. Most state-funded laboratories are housed in buildings over 40 years old and are in urgent need of renovation. The physical layout of many laboratories is not conducive to efficient workflow or effective access control. Climate control is often absent, and much of the furniture is as old as the buildings themselves. However, most laboratories have established reliable systems for medical waste collection and decontamination.

## Biorisk management

Scores for biosafety and biosecurity were low: 35% for SESS and 26% for CDL laboratories. Biosafety and biosecurity in Kyrgyz laboratories have been governed by outdated norms. While national laboratories handling high-risk pathogens (e.g., Centers for TB, HIV, Quarantine Infections and Especially Dangerous Pathogens) apply international standards,

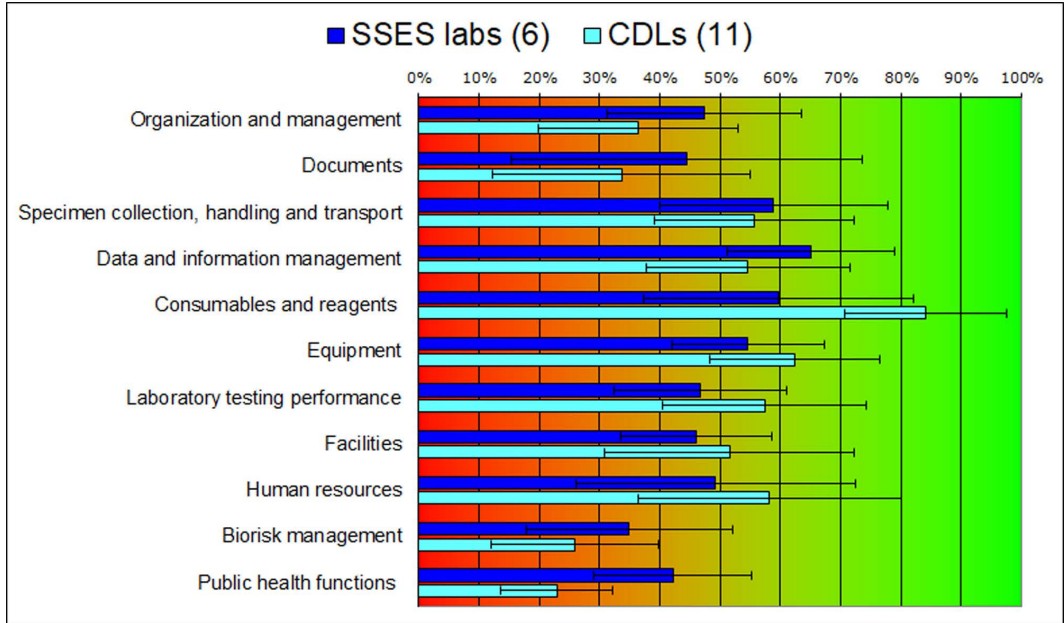

**Fig 1. Average LAT scores for 17 laboratories.** CDL – clinical diagnostic laboratory; SSES – State Sanitary and Epidemiological Surveillance. Note: Score below 50% means that this element requires significant improvement, between 50 and 85% means improvement is necessary, and above 85% means good standing. Whiskers denote standard deviation.

most routine SESS and CDL laboratories lack basic biosafety measures. Common issues include the absence of access control, poor ventilation, limited use of personal protective equipment, and no inventory management for hazardous materials. Nevertheless, many laboratories are equipped with autoclaves to disinfect contaminated waste, and autoclaves are regularly certified. Disinfectants and related protocols also are generally available.

### Public health functions

This was one of the weakest areas assessed. CDL laboratories rarely collaborate with each other or participate in public health programs and emergency preparedness, reflected in their average score of 23%. The LAT score for this component evaluates laboratories' roles in disease surveillance, data reporting, and participation in healthcare initiatives. While some regional and national SESS laboratories monitor specific infections and follow reporting protocols, these efforts are limited to a small number of pathogens. Important threats such as antimicrobial resistance (AMR) are generally not monitored. Only three SESS laboratories scored above 50% in this category, while peripheral SESS laboratories scored below 35%.

### Specimen collection, handling and transportation

Scores for this component averaged 59% for SESS and 56% for CDL laboratories. There was substantial variability among individual laboratories, with some achieving scores between 65–79%, while others scored as low as 19%. Common issues included the absence of written procedures, frequent errors in documentation accompanying specimens, and a lack of suitable containers and transportation systems.

### Data and information management

This area encompasses reporting of test results, management of patient data confidentiality, and the use of digital systems. SESS and CDL laboratories scored 65% and 55% respectively, indicating moderate implementation of these practices.

### Equipment

SESS laboratories scored 55%, and CDLs 62% in this component, which includes equipment inventory and maintenance. Much of the equipment is 20–30 years old, with quality often depending on laboratory leadership and donor-funded initiatives. While routine maintenance is performed for basic equipment like autoclaves, thermometers, and scales, there is a shortage of specialists for servicing complex instruments (e.g., PCR machines, automated analyzers). Most laboratories have limited or outdated computer equipment.

### Laboratory testing performance

None of the 17 assessed laboratories are accredited to any ISO quality standard. Although a national laboratory quality standard was developed in 2018 with WHO support, it remains unapproved by the MOH. As such, no laboratory is currently licensed or accredited. This deficiency is reflected in the low LAT scores for this component – 47% for SESS and 57% for CDLs. These scores consider the use of control materials, presence of standard operating procedures (SOPs), adequacy of reagents and equipment, staff qualifications, and participation in EQA programs.

Only 13% of CDL tests from the project list are covered by EQA (see Table 3). National and subnational reference CDLs are not currently EQA providers. For laboratories, participation in EQA is now voluntary. Some CDL laboratories in Bishkek participate in EQA provided by the national Association of CDL Professionals, but EQA participation is limited by the absence of a national sample transportation system. In 2021, only three of 11 CDLs in Bishkek participated in an EQA scheme, while two future reference laboratories were not assessed.

A small number of SESS laboratories (2–5) participate in the United Kingdom National EQA Schemes (UKNEQAS) programs for AMR-related tests under WHO support. However, results are rarely analyzed or used for improvement. The national reference SESS laboratory provides EQA for some bacteriology tests within the SESS network, but detailed outcomes and analyses were not available.

### Human resources

This component assesses staffing adequacy, qualifications, and access to continuing education. Scores were 48% for SESS and 59% for CDL laboratories, indicating notable gaps in workforce capacity and development.

### Consumables and reagents

This component received the highest average scores among all the assessed areas. A centralized procurement system and market restrictions on registered products have been established in the country and are generally functioning well. Internal financial oversight within organizations supports effective inventory tracking. However, delivery delays remain a commonly reported issue.

## Discussion

The assessment findings reveal significant variation in laboratory service capacity, testing scope, and quality performance across the country. Facilities with the highest LAT scores and testing capacity are concentrated in urban centers – particularly Bishkek – where infrastructure, staffing levels, and supply chains are more robust. These laboratories are also more likely to participate in EQA schemes and deliver a broader range of essential diagnostics.

In contrast, many peripheral laboratories, particularly those attached to outpatient clinics in rural or semi-urban areas, operate under considerable constraints. These include outdated infrastructure, manual testing methods, minimal range of tests, and no national EQA participation. Despite similar operating costs, these laboratories are underutilized and unable to consistently deliver quality-assured diagnostics.

The performance variability observed is not incidental, but systemic. The gaps identified in each facility reflect structural and resource limitations that are replicated across many laboratories nationwide. Attempting to strengthen all existing laboratories to a high standard would be both financially unsustainable and operationally impractical. Instead, the findings support a more strategic reform: centralizing laboratory services to improve efficiency, ensure quality, and enable equitable access to essential diagnostics.

The rationale for laboratory centralization is grounded in several key findings from the assessment:

- Redundancy and inefficiency of low-capacity laboratories: Many peripheral laboratories perform limited testing, with low throughput and no participation in quality programs, yet require comparable human and financial resources.

- Clustering of functional capacity and quality participation: The best-performing laboratories – with the highest LAT scores and participation in EQA – are already concentrated in urban centers.

- Potential for scale and cost-effectiveness: Centralized laboratories can maximize use of automated equipment and trained personnel, support broader quality assurance coverage, and reduce duplication of services.

This interpretation of findings has led to the development of a tiered laboratory optimization framework. The framework defines laboratories by service level – national, subnational, inter-district, express laboratories, and sample collection points (SCPs) – and aligns test profile with WHO Essential Diagnostic List recommendations and national health priorities. Key components of the framework include:

**Centralization of laboratory services.** In shaping the optimization framework, a network of centralized laboratories will be first assigned accordingly. Most existing outpatient CDLs will be closed or repurposed into SCPs. Future reference laboratories at the national and subnational levels will handle the majority of diagnostics, while inter-district and express laboratories will provide rapid turnaround for urgent cases. This model is expected to ensure wider coverage and accessibility to higher quality tests through fewer but better-equipped facilities. Existing staff of small laboratories will be retrained and redeployed to central laboratories or retained in SCPs to support sample referral process.

**Facilities and equipment upgrades.** All 17 designated laboratories will be renovated or expanded to meet biosafety and quality requirements. Equipment lists tailored to test profile and service levels have been developed to ensure functional readiness.

**Test standardization.** The national nomenclature, currently containing over 1,000 test types, has been streamlined to 61 essential clinically relevant tests [26], and assigned to laboratories according to service level.

**Strengthening laboratory networks.** To improve national diagnostic capacity, over 100 SCPs will be linked to the 17 laboratories through a coordinated sample referral and transportation system supported by a laboratory information management system (LIMS). To optimize available resources, a small-scale outsourcing of specialized or low-volume tests to the private sector will be selectively explored. Upgrades in infrastructure, including hardware, equipment, and computerization, will be implemented alongside improvements in management of the laboratories such as costing, budgeting, workload planning, and networking.

**Training and continuous education.** Recognizing the need for a skilled workforce, the project will focus on capacity building through continuing education programs. A review of existing training institutes and a needs assessment will guide the development of curricula in key areas such as advanced equipment operation, modern diagnostic methods, QMS, biorisk management, and sample transportation. Training will be tailored to different experience levels, offering beginner, advanced, and refresher courses in topics such as vaccinology and antimicrobial resistance. Reference laboratories will also function as training centers (and skill laboratories) providing hands-on education. Staff of reference laboratories will be encouraged to contribute as part-time instructors, leveraging their practical expertise.

 

**Advancing quality and accreditation.** To ensure high standards, reference laboratories will be supported in achieving ISO 15189 accreditation, while other laboratories will work toward compliance with national quality and safety standards. The existing regulatory framework governing health laboratories will be reviewed to identify gaps for improvement. The most suitable mechanism – licensing, certification, or accreditation – will be established in collaboration with CLC, MOH, Kyrgyz Center for Accreditation (KCA), and WHO to ensure compliance of laboratories with the national standards.

Support will be provided for the development of EQA programs, including preparation of sample panels, logistics, communication with participants, collection and analysis of results, software, and websites. The reference laboratories will additionally prepare for ISO 17043 (general requirements for the competence of proficiency testing providers) accreditation as EQA providers.

**Costing and sustainability analysis**. Adequate and sustainable funding for clinical laboratories is key to ensuring the sustainability of laboratory services. The assessment revealed considerable variation in cost per test, indicating inefficiencies in resource allocation and posing a challenge in advocating budget allocation that matches the actual cost of the tests. A detailed costing review – including the costs of human resources, infrastructure, equipment, supplies, and quality and safety systems – will inform sustainable budget planning for the future laboratory network.

Experience from high income settings has shown that centralization of lab services might not be well-received by clinicians. Arguably, taking away onsite laboratory services from hospitals might risk, for example, delayed responses to clinical problems [27]. However, given the current state of most clinical diagnostic laboratories in the Kyrgyz Republic, the proposed centralization strategy is a direct response to the performance disparities and inefficiencies highlighted by the assessment. It aims to ensure access to reliable, quality-assured diagnostics nationwide while building a more resilient, cost-effective laboratory system. If implemented as planned, this model could serve as a reference for other low- and middle-income countries seeking to modernize and optimize fragmented laboratory systems.

## Conclusion

By optimizing healthcare laboratories, the *Strengthening Regional Health Security Project* led by the MOH of the Kyrgyz Republic aims to improve the population's access to quality essential laboratory tests and address the five greatest weaknesses identified by WHO's Laboratory Assessment Tool: (i) weak organization and management of laboratory, (ii) insufficient documentation management, (iii) inadequate facilities for biosafety, (iv) lack of biorisk management, and (v) unmet public health functions. The project will be implemented in two major cities and two regions of the country. The government will establish a laboratory network consisting of 6 public health laboratories and 11 clinical diagnostic laboratories in Chui and Osh regions, which will be upgraded and equipped based on modern quality and safety standards. Of these 17 laboratories, four key laboratories in Bishkek and Osh cities will be further strengthened into the country's leading national and subnational reference laboratories with an aim to obtain ISO accreditation. In addition, around 100 lower-level laboratories of the two regions will be reprofiled into sample collection points via sample transportation and laboratory information management system. Continuous quality improvement of laboratory networks will be achieved through strengthening governance capacity for the national laboratory system including regulation, standards, planning, financing, management, monitoring, and studies for innovative solutions.

Periodic project monitoring and process evaluation of laboratory optimization reform will be undertaken to document implementation experience, good practices, and lessons learned, which may be informative for other settings considering similar reform.

## Acknowledgments

The authors acknowledge the Ministry of Health of the Kyrgyz Republic, members of the Coordination Laboratory Council, Sergejs Nikišins, Joanna Zwetyenga, and Mustafa Aboualy for sharing their valuable insights.

## Author contributions

**Conceptualization:** Taalaigul Sabyrbekova, Elmira Turkmenova, Brian Chin.

**Data curation:** Ping Ling Yeoh, Olga Slobodskaya.

**Investigation:** Taalaigul Sabyrbekova, Elmira Turkmenova, Ping Ling Yeoh, Olga Slobodskaya.

**Project administration:** Cebele Wong, Brian Chin.

**Supervision:** Brian Chin.

**Writing – original draft:** Taalaigul Sabyrbekova, Ping Ling Yeoh, Olga Slobodskaya.

**Writing – review & editing:** Taalaigul Sabyrbekova, Elmira Turkmenova, Ping Ling Yeoh, Olga Slobodskaya, Cebele Wong, Brian Chin.

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
