## [Decision Letter · Decision Letter 0]

PGPH-D-24-03057

Assessment and optimization plan for enhancing public medical laboratory services in the Kyrgyz Republic

Dear Dr. Sabyrbekova,

Thank you for submitting your manuscript to PLOS Global Public Health. After careful consideration, we feel that it has merit but does not fully meet PLOS Global Public Health’s publication criteria as it currently stands. Therefore, we invite you to submit a revised version of the manuscript that addresses the points raised during the review process.

The manuscript has been evaluated by two reviewers, and their comments are available below and in the attached document.

Could you please carefully revise the manuscript to address all comments raised?

We look forward to receiving your revised manuscript.

Kind regards,

Steve Zimmerman, PhD

PLOS Staff Editor

Journal Requirements:

Additional Editor Comments (if provided):

Reviewers' comments:

Reviewer's Responses to Questions

**Comments to the Author**

1. Does this manuscript meet PLOS Global Public Health’s publication criteria?

Reviewer #1: Partly

Reviewer #2: Partly

2. Has the statistical analysis been performed appropriately and rigorously?

Reviewer #1: N/A

Reviewer #2: N/A

3. Have the authors made all data underlying the findings in their manuscript fully available (please refer to the Data Availability Statement at the start of the manuscript PDF file)?

Reviewer #1: Yes

Reviewer #2: No

4. Is the manuscript presented in an intelligible fashion and written in standard English?

Reviewer #1: Yes

Reviewer #2: No

Reviewer #1: General comments:

This manuscript describes an assessment of the laboratory network in the Kyrgyz Republic, findings on key gaps, and a set of recommendations for improvement and optimization. The analysis is based on the application of a WHO assessment tool and the findings are reported. Because it is national in nature and multiple dimensions were evaluated, this assessment is a valuable contribution the literature on laboratory strengthening.

Overall, the manuscript would benefit from a clearer description of the LAT findings in the text and of how these drive the recommendations and conclusions.

Specific comments:

The Introduction provides a good description of the structure of the national laboratory network and the history of development efforts. The Introduction is, however, quite lengthy and would benefit from editing to more efficiently describe the necessary background and rationale for the study. There are two paragraphs on Page 3 that describe how laboratories are financed – these might be less relevant for the Introduction, or they could be shortened significantly to distil the main relevant points.

Page 5: “Another way to assess the quality of laboratory services was to evaluate the range (list) of laboratory tests offered and whether the accuracy of test results was confirmed by a qualified independent external quality assessment (EQA) program.” Was this method also used on the study?

The Results section would benefit from a more quantitative description of the findings as summarized in Figure 1 and Tables 2 and 3. It would be useful to more closely structure the study findings around these data, highlighting key points and differences that drive the conclusions and the decisions around optimization of the network. For example, it is not clear what drives the decision to optimize/centralize testing described in the Discussion on Page 12.

The first two paragraphs of the Results section seem to be background material already known before the study, and not findings of the study. Perhaps they can be summarized and shifted to related sections on the Introduction or Methods sections.

While data on the lab finances are collected, it is not clear how these are used in the analysis and what the recommendations are.

Reviewer #2: My main observations about this article

A professional English language edit is required.

It was not properly structured as an original full length (research) article.

There are sections of the methodology that were not clear

The results section contains more information than results...some methodology, and a lot of discussion of results

The discussion section is a list of recommendations rather than a discussion of the results.

Some more specific comments are in the attached document.

**Do you want your identity to be public for this peer review?** For information about this choice, including consent withdrawal, please see our Privacy Policy

Reviewer #1: No

Reviewer #2: No

---

## [Decision Letter · Decision Letter 1]

PGPH-D-24-03057R1

Assessment and optimization plan for enhancing public medical laboratory services in the Kyrgyz Republic

Dear Dr. Sabyrbekova,

Thank you for submitting your manuscript to PLOS Global Public Health. After careful consideration, we feel that it has merit but does not fully meet PLOS Global Public Health’s publication criteria as it currently stands. Therefore, we invite you to submit a revised version of the manuscript that addresses the points raised during the review process.

The manuscript has been evaluated by two reviewers, and their comments are available below. 

Although both reviewers are satisfied with the revisions following their previous comments, reviewer 2 has a few more requests for minor changes, which are specified in the attached file.

Could you please carefully revise the manuscript to address all comments raised?

We look forward to receiving your revised manuscript.

Kind regards,

Steve Zimmerman, PhD

PLOS Staff Editor

Journal Requirements:

1. Please include the following request in the decision letter, and ping me with follow-up. “Please include a complete copy of PLOS’ questionnaire on inclusivity in global research in your revised manuscript. Our policy for research in this area aims to improve transparency in the reporting of research performed outside of researchers’ own country or community. The policy applies to researchers who have travelled to a different country to conduct research, research with Indigenous populations or their lands, and research on cultural artefacts. The questionnaire can also be requested at the journal’s discretion for any other submissions, even if these conditions are not met. Please find more information on the policy and a link to download a blank copy of the questionnaire here: https://journals.plos.org/globalpublichealth/s/best-practices-in-research-reporting. Please upload a completed version of your questionnaire as Supporting Information when you resubmit your manuscript.

Additional Editor Comments (if provided):

Reviewers' comments:

Reviewer's Responses to Questions

**Comments to the Author**

Reviewer #1: All comments have been addressed

Reviewer #2: All comments have been addressed

publication criteria?

Reviewer #1: Yes

Reviewer #2: Yes

3. Has the statistical analysis been performed appropriately and rigorously?

Reviewer #1: Yes

Reviewer #2: N/A

4. Have the authors made all data underlying the findings in their manuscript fully available (please refer to the Data Availability Statement at the start of the manuscript PDF file)?

Reviewer #1: Yes

Reviewer #2: Yes

5. Is the manuscript presented in an intelligible fashion and written in standard English?

Reviewer #1: Yes

Reviewer #2: Yes

Reviewer #1: (No Response)

Reviewer #2: The authors have addressed the comments in the earlier review.

There are additional minor issues to be addressed in the attached document

**Do you want your identity to be public for this peer review?** For information about this choice, including consent withdrawal, please see our Privacy Policy

Reviewer #1: No

Reviewer #2: No

---

## [Editor Report · Decision Letter 2]

Assessment and optimization plan for enhancing public medical laboratory services in the Kyrgyz Republic

PGPH-D-24-03057R2

Dear Mrs. Sabyrbekova,

We are pleased to inform you that your manuscript 'Assessment and optimization plan for enhancing public medical laboratory services in the Kyrgyz Republic' has been provisionally accepted for publication in PLOS Global Public Health.

Best regards,

Julia Robinson

Executive Editor